# Immunodetection of P2X2 Receptor in Enteric Nervous System Neurons of the Small Intestine of Pigs

**DOI:** 10.3390/ani12243576

**Published:** 2022-12-17

**Authors:** Sylwia Mozel, Marcin B. Arciszewski

**Affiliations:** Department of Animal Anatomy and Histology, Faculty of Veterinary Medicine, University of Life Sciences, 12 Akademicka St., 20-950 Lublin, Poland

**Keywords:** P2X2, galanin, substance P, vasoactive intestinal polypeptide, enteric nervous system, small intestine, pig

## Abstract

**Simple Summary:**

The enteric nervous system (ENS) is responsible for regulating the main functions of the intestines, i.e., absorption, secretion, peristalsis and sensation. The ENS is composed of nerve cells that form clusters in the form of ganglia arranged in nerve plexuses. Individual ganglia communicate through a system of nerve fibres. Each nerve cell or nerve fibre contains biologically active substances exerting a specific effect on the ENS. One such substance is extracellular adenosine 5′-triphosphate (ATP), which by activating purinoreceptors (including P2X2 receptors) may affect the functioning of the ENS. Previous research on the P2X2 receptor has focused mainly on laboratory animals, while little is known about this receptor in large mammals. Therefore, our research concerned the presence of the P2X2 receptor in the ENS of the small intestine of the pig and whether it coexists with the biologically active substances included in standard analyses in the ENS. The research showed that the P2X2 receptor is present in the neurons of the small intestine and coexists with some of the biologically active substances tested. These findings may suggest that the P2X2 receptor, activated by ATP, may exert an influence on the function of the small intestinal ENS of the pig.

**Abstract:**

Extracellular adenosine 5′-triphosphate (ATP) is one of the best-known and frequently studied neurotransmitters. Its broad spectrum of biological activity is conditioned by the activation of purinergic receptors, including the P2X2 receptor. The P2X2 receptor is present in the central and peripheral nervous system of many species, including laboratory animals, domestic animals, and primates. However, the distribution of the P2X2 receptor in the nervous system of the domestic pig, a species increasingly used as an experimental model, is as yet unknown. Therefore, this study aimed to determine the presence of the P2X2 receptor in the neurons of the enteric nervous system (ENS) of the pig small intestine (duodenum, jejunum, and ileum) by the immunofluorescence method. In addition, the chemical code of P2X2-immunoreactive (IR) ENS neurons of the porcine small intestine was analysed by determining the coexistence of selected neuropeptides, i.e., vasoactive intestinal polypeptide (VIP), substance P (sP), and galanin. P2X2-IR neurons were present in the myenteric plexus (MP), outer submucosal plexus (OSP), and inner submucosal plexus (ISP) of all sections of the small intestine (duodenum, jejunum, and ileum). From 44.78 ± 2.24% (duodenum) to 63.74 ± 2.67% (ileum) of MP neurons were P2X2-IR. The corresponding ranges in the OSP ranged from 44.84 ± 1.43% (in the duodenum) to 53.53 ± 1.21% (in the jejunum), and in the ISP, from 53.10 ± 0.97% (duodenum) to 60.57 ± 2.24% (ileum). Immunofluorescence staining revealed the presence of P2X2-IR/galanin-IR and P2X2-IR/VIP-IR neurons in the MP, OSP, and ISP of the sections of the small intestine. The presence of sP was not detected in the P2X2-IR neurons of any ganglia tested in the ENS. Our results indicate for the first time that the P2X2 receptor is present in the MP, ISP, and OSP neurons of all small intestinal segments in pigs, which may suggest that its activation influences the action of the small intestine. Moreover, there is a likely functional interaction between P2X2 receptors and galanin or VIP, but not sP, in the ENS of the porcine small intestine.

## 1. Introduction

Extracellular adenosine 5′-triphosphate (ATP) is one of the most frequently studied neurotransmitters and neuromodulators, present in a wide population of various neurons types of both the central (CNS) and peripheral (PNS) nervous system [1,2]. The action of ATP under both physiological and pathophysiological conditions is associated with the activation of purinergic receptors, which include P2X receptors [3,4]. Seven subtypes have thus far been identified in the P2X family (P2X1–P2X7), of which one of the most common (in the nervous system) is the P2X2 receptor [1,2]. Immunoexpression of the P2X2 receptor has been demonstrated in the CNS, e.g., in the hippocampus, cortex, amygdala, thalamus, and hypothalamus [5]. Moreover, the presence of P2X2 receptors has been demonstrated in the PNS, including the enteric nervous system (ENS) [6].

Previous reports have indicated that the ENS is responsible for the regulation of key functions of the gastrointestinal tract (GIT), including absorption, secretion, peristalsis, or transmission of sensation and pain [7]. In most laboratory animals, the ENS (within the intestines) is composed of the myenteric plexus (MP) and submucosal plexus (SP) [8,9,10,11]. However, the ENS structure of pig intestines is slightly different, because the submucosal plexus is divided into the inner submucosal plexus (ISP) and outer submucosal plexus (OSP) [12]. Noteworthy, similar architecture of SP forming a kind of two-layered interconnected neuronal network seems to be present also in other animals’ species including horse, dog or monkey [13].

The ENS is composed of several types of neurons: interneurons, motor neurons, intrinsic primary afferent neurons (IPAN), and intestino-fugal afferent neurons (IFAN) [14]. Each type of neuron can be defined by a specific set of neurotransmitters or neuropeptides that potentially allows these cells to be assigned a probable function [14]. The set of biologically active substances in individual types of ENS neurons may be slightly different in different species, which suggests that the neuronal mechanisms regulating the activity of the GIT are species-dependent [15,16].

The vast majority of studies on the immunoexpression of the P2X2 receptor in the ENS have concerned laboratory animals (rats, mice, or guinea pigs), while reports on large domestic mammals are scarce, which is an obvious gap in neuroanatomical research [6,17,18]. Therefore, the present study aimed to determine the degree of immunoexpression of the P2X2 receptor and to demonstrate whether the P2X2 receptor is associated with galanin-immunoreactive (IR), substance P-IR (sP), and vasoactive intestinal polypeptide (VIP-IR) neurons in the ENS of the small intestine of pigs. The selection of galanin, sP and VIP was based on the previous findings showing their neurotransmittery role in certain subpopulations of enteric neurons (including sensory and secretomotor) as well as a potential neuroprotective role in the gut disorders with the pain symptoms.

## 2. Materials and Methods

### 2.1. Animals and Tissue Processing

All experimental procedures were approved by the Second Local Ethics Committee at the University of Life Sciences in Lublin, Poland (decision no. 30/2015 of 26 May 2015). This study was conducted on five (*n* = 5) 8-week-old Landrace × Pietrain crossbred pigs of both sexes (three males and two females), weighing 15–20 kg and in good physical condition. All animals were housed individually with access to water and standard feed.

Each animal was premedicated by intramuscular injection of azaperone (Stresnil, Janssen Pharmaceutica N.V., Belgium; dose: 2 mg/kg b.w.). The animals were then euthanized by intravenous injection of a lethal dose of sodium pentobarbitone (Morbital, Biowet Puławy, Poland; dose: 50 mg/kg b.w.). Directly after euthanasia, the guts were accessed by incision of the abdominal wall along the linea alba. After visualizing the position of each part of the small intestine (duodenum, jejunum, and ileum), sections approximately 2.5 cm long were collected and carefully rinsed in saline solution (37 °C). Next, the sections were stretched and pinned on pieces of balsa wood and fixed with Stefanini’s solution (containing paraformaldehyde and picric acid) for 24 h. Then, the material was placed in a container filled with 18% sucrose solution (4 °C) with 0.01% sodium azide (Avantor Performance Materials Poland S.A., Gliwice, Poland). The samples were rinsed in the sucrose solution (which was replaced with fresh solution every day) for several days until the material fell to the bottom of the container. Then, the samples were embedded in Tissue-Tek^®^ O.C.T.™ Compound (Sakura Finetek USA, Inc., Torrance, CA, USA), and frozen sections 10 µm thick were made using a cryostat (HM525NX, Thermo Scientific, Waltham, MA, USA). All frozen sections were placed on adhesion glass slides (Superfrost^®^ Plus, Menzel-Gläser, Thermo Scientific, Braunschweig, Germany) and stored (at −70 °C) for further use.

### 2.2. Immunofluorescence

One of every three frozen sections was stained according to the following immunohistochemical method. After the defrosting of selected slides, the sections were outlined with a hydrophobic marker (ImmEdge™ Hydrophobic Barrier Pen, Vector Laboratories, Burlingame, CA, USA) and washed (3 × 15 min) at room temperature (RT) in 0.01 M phosphate-buffered saline (PBS; pH = 7.4) with 0.25% Triton X-100 (Sigma-Aldrich, Saint Louis, MO, USA). Next, the sections were placed in a dark humid chamber, covered with a mixture of primary antibodies, and incubated overnight (RT). To visualize P2X2-IR neurons, anti-Hu C/D antibodies were combined with P2X2 antibodies. For co-localization studies P2X2 antibodies were combined with either VIP, galanin or sP antibodies. To detect antigen–antibody complexes (immediately after incubation with primary antibodies), the sections were washed (3 × 15 min) in PBS and placed in a dark humid chamber, where they were incubated in a solution containing species-specific secondary antibodies conjugated with fluorochromes for 1 h at RT (Alexa Fluor 488, Alexa Fluor 594). The specifications and dilutions of the primary and secondary antibodies used are listed in Table 1. Following incubation with secondary antibodies, the slides were once again washed in PBS (3 × 15 min, RT). In the final step, the slides were mounted in phosphate-buffered glycerol (pH = 8.2) and coverslipped. Antibodies specificity was verified using two different control procedures. The first control test involved immunofluorescence staining (according to the method described above), in which the primary antibodies were omitted or replaced with non-IR goat sera. In the second control procedure (pre-absorption experiment), primary antibodies were inactivated by adding an excess of the target synthetic protein. No immunoreaction was observed in any of the control sections.

### 2.3. Cell Counting, Imaging, and Statistical Analysis

The stained sections were viewed under an epifluorescence microscope (BX-61 Olympus, Nagano, Japan) connected to a digital camera (C11440-36U, Hamamatsu Photonics, Shizuoka, Japan), equipped with appropriate filters for detection of Alexa Fluor 488 (470–490 nm; MNIBA2, Olympus, Japan) and Alexa Fluor 594 (545–580 nm; MWIY2, Olympus, Japan). The images were captured using Cell^M software (Olympus cellSense Standard, Japan). The morphometric analysis was performed using ImageJ 1.52 software (NIH, Bethesda, MD, USA) and included determination of single ganglion area, number of neurons per ganglion, and neuron diameter. From each animal (*n* = 5), not less than 300 neurons of the MP, OSP, and ISP of the duodenum, jejunum, and ileum were examined. The percentage of neurons expressing the P2X2 receptor was calculated from the total number of all nerve cells labelled with the Hu C/D pan-neuronal marker. The percentages of P2X2-positive neurons showing co-localization with the biologically active substances tested were calculated and expressed as a percentage of the total number of P2X2-IR neurons. The data were analysed using Statistica 13.3 software (TIBCO Software Inc., 2017; Palo Alto, CA, USA). One-way analysis of variance (ANOVA) and Tukey’s honestly significant difference (HSD) test were performed. All differences were considered significant at *p* < 0.05. The data are expressed as arithmetic mean ± standard deviation (S.D.) or percentage ± S.D.

## 3. Results

### 3.1. Hu C/D/P2X2

The immunohistochemical analyses using P2X2 antisera and Hu C/D revealed the presence of P2X2-positive neurons in all segments of the small intestine (duodenum, jejunum, and ileum). Neurons showing co-localization of Hu C/D and P2X2 were found in the MP, OSP, and ISP. However, the percentages of Hu C/D-IR/P2X2-IR neurons in a given plexus (MP, ISP, and OSP) varied depending on the section of the small intestine. Detailed percentages of the subpopulations of MP, ISP, and OSP neurons co-immunoexpressing Hu C/D and P2X2 are presented in Figure 1.

In the MP of the duodenum, Hu C/D-IR/P2X2-IR neurons constituted 44.78 ± 2.24% of nerve cells (Figure 1). In a single MP ganglion (mean area 2826.15 ± 149.87 µm^2^; morphometrical data are summarized in Table 2) of the duodenum, about 10.54 ± 3.50 neurons were Hu C/D-IR, of which 4.26 ± 1.95 nerve cells co-immunoexpressed P2X2. The average diameter of Hu C/D-IR/P2X2-IR neurons in the duodenal MP was 20.18 ± 3.11 µm. Among duodenal OSP nerve cells, 44.84 ± 1.43% were simultaneously IR for P2X2 as well as Hu C/D. The average diameter of Hu C/D-positive/P2X2-positive neurons in the duodenal OSP was 14.37 ± 2.63 µm. A single OSP ganglion of the duodenum was composed of 8.59 ± 5.26 neurons, but only 4.82 ± 1.87 of nerve cells were Hu C/D-IR/P2X2-IR. The mean area of a single OSP ganglion was 2285.57 ± 177.42 µm^2^. In the population of duodenal ISP cells, 53.10 ± 0.97% of neurons were Hu C/D-positive/P2X2-positive, with an average diameter of 14.73 ± 2.22 µm. A single ISP ganglion contained 7.09 ± 3.37 neurons, of which 4.11 ± 1.26 co-immunoexpressed both Hu C/D and P2X2. The mean area of an ISP ganglion of the duodenum with Hu C/D-IR/P2X2-IR neurons was 1711.00 ± 103.82 µm^2^.

Among MP neurons of the jejunum, 61.07 ± 2.16% were Hu C/D-IR/P2X2-IR, and the average diameter of a single such cell was 21.24 ± 1.10 µm (Figure 2A). The mean area of a single duodenal MP ganglion was 4643.7 ± 335.91 µm^2^ and contained 8.10 ± 4.60 Hu C/D-IR neurons. In a single duodenal MP ganglion, 4.40 ± 1.70 neurons co-immunoexpressed Hu C/D and P2X2. A single OSP ganglion in the jejunum contained 9.11 ± 3.80 neurons, of which 5.32 ± 2.25 nerve cells were Hu C/D-IR/P2X2-IR (Figure 2B). The mean area of a single jejunal OSP ganglion was 1970.24 ± 172.62 µm^2^. The percentage of nerve cells with P2X2 and Hu C/D immunoreaction in a jejunal OSP was 53.53 ± 1.21%. The mean diameter of neurons located in this plexus was 17.91 ± 1.12 µm. In the ISP of the jejunum, 57.85 ± 1.12% of neurons were found to be Hu C/D-IR/P2X2-IR (with a mean diameter of 15.82 ± 1.04 µm; Figure 2C). The average area of a single ISP ganglion of this part of the small intestine was 1953.9 ± 142.23 µm^2^. These ganglia contained 10.21 ± 4.59 neurons, of which 6.64 ± 1.65 exhibited Hu C/D and P2X2 immunoexpression.

The percentage of Hu C/D-IR/P2X2-IR neurons in the MP of the ileum was 63.74 ± 2.67%, and the mean diameter of these cells was 20.62 ± 1.53 µm. A single ileal MP ganglion (mean area 4449.81 ± 98.85 µm^2^) was composed of 9.22 ± 3.24 nerve cells, but only 6.52 ± 2.31 of these were Hu C/D-positive/P2X2-positive. In the OSP of the ileum, 52.62 ± 3.87% of neurons were Hu C/D-IR/P2X2-IR, with a mean diameter of 18.52 ± 1.30 µm. The mean area of a single OSP ganglion was 2175.18 ± 476.44 µm^2^, and it contained 12.77 ± 3.68 neurons, of which 6.71 ± 1.82 were Hu C/D-IR/P2X2-IR. In the ISP of the ileum, the average area of a single ganglion was 2239.73 ± 391.07 µm^2^, and it contained 13.80 ± 4.12 neurons. A single such ganglion contained on average 8.87 ± 2.13 Hu C/D-IR/P2X2-IR neurons. Among ileal ISP neurons, 60.57 ± 2.24% co-immunoexpressed Hu C/D and P2X2, and the mean diameter of a single nerve cell was 15.03 ± 0.97 µm.

There were no statistical differences (*p* < 0.05) in the number of Hu C/D-IR (*n* = 5) neurons per ganglion in analogous plexuses (MP vs. MP, OSP vs. OSP, ISP vs. ISP) between sections of the small intestine (duodenum, jejunum, and ileum). Statistical differences (*p <* 0.05) in the percentages of Hu C/D-IR/P2X2-IR neurons (*n* = 5) were found between analogous plexuses (MP vs. MP, OSP vs. OSP, ISP vs. ISP) in different segments of the small intestine (Figure 1). In addition, statistical differences in the mean area of a single MP ganglion containing Hu C/D-IR/P2X2-IR neurons (*p* < 0.05) were shown between the duodenum and the jejunum and between the duodenum and ileum. In contrast, comparison of the mean areas of a single OSP ganglion containing Hu C/D-IR/P2X2-IR neurons showed no statistically significant differences (*p* < 0.05; duodenum vs. jejunum; duodenum vs. ileum; jejunum vs. ileum). In the ISP, statistically significant differences (*p* < 0.05) in the mean area of a single ganglion (containing Hu C/D-positive/P2X2-positive nerve cells) were shown only between the duodenum and the ileum. Comparison of the average number of Hu C/D-IR/P2X2-IR neurons per MP ganglion revealed no statistically significant differences (*p* < 0.05) between sections of the small intestine (duodenum, jejunum, and ileum). Moreover, no statistically significant differences *(p* < 0.05) were found in the mean (*n* = 5) number of Hu C/D-IR/P2X2-IR neurons per OSP ganglion between sections of the small intestine. In the ISP, however, statistically significant differences (*p* < 0.05) in the mean number of Hu C/D-IR/P2X2-IR neurons (*n* = 5) per ganglion were found only between the duodenum and jejunum. There were no statistically significant differences (*p* < 0.05) between the mean diameters of Hu C/D-IR/P2X2-IR neurons of either the MP or ISP between sections of the small intestine. Statistically significant differences (*p* < 0.05) in the mean diameter of Hu C/D-positive/P2X2-positive neurons were found only in the OSP between the duodenum and jejunum and between the duodenum and ileum.

### 3.2. P2X2/Galanin

In the MP of the duodenum, 21.39 ± 2.18% of neurons were simultaneously IR for P2X2 and galanin (Figure 3A). The mean diameter of the P2X2-IR/galanin-IR neurons of the duodenal MP ganglia was 20.25 ± 2.12 µm. In a single duodenal MP ganglion, the mean number of nerve cells immunoexpressing both P2X2 and galanin was 2.19 ± 1.03. On average 3.08 ± 1.39 P2X2-positive/galanin-positive neurons were located in a single duodenal OSP ganglion, and their mean diameter was 14.11 ± 2.60 µm (Figure 3C). In the OSP of the duodenum, on average 25.93 ± 0.92% of neurons were P2X2-IR/galanin-IR. Among neurons in the ISP of the duodenum, 27.32 ± 1.48% immunoexpressed P2X2 and galanin simultaneously, and their mean diameter was 14.21 ± 2.21 µm (Figure 3B). There were 2.79 ± 1.23 P2X2-positive/galanin-positive neurons per duodenal ISP ganglion.

The MP of the jejunum contained 25.15 ± 4.48% P2X2-IR/galanin-IR nerve cells. The mean diameter of these neurons was 17.20 ± 2.18 µm. There were 3.31 ± 1.23 P2X2-positive/galanin-positive neurons present in a single MP ganglion of the jejunum. The OSP of the jejunum contained on average 31.16 ± 3.44% P2X2-IR/galanin-IR neurons, while a single ganglion contained 2.44 ± 1.61 neurons. The mean diameter of P2X2-positive/galanin-positive neurons in the OSP of the jejunum was 14.84 ± 1.69 µm. In the ISP of the jejunum, 32.19 ± 3.24% of neurons were P2X2-IR/galanin-IR, and a single ISP ganglion contained 2.91 ± 1.04 of these neurons. The average diameter of the P2X2-positive/galanin-positive neurons was 15.34 ± 1.27 µm.

Among nerve cells of the ileal MP, 23.69 ± 2.76% showed simultaneous immunoexpression of P2X2 and galanin, and their mean diameter was 17.14 ± 1.54 µm. A single MP ganglion of the ileum contained 2.79 ± 1.36 P2X2-IR/galanin-IR neurons. A single OSP ganglion of the ileum contained 2.58 ± 1.07 P2X2-positive/galanin-positive neurons, with a mean diameter of 15.98 ± 3.70 µm. The percentage of P2X2-IR/galanin-IR neurons was 32.52 ± 3.62% in the OSP of the ileum and 32.07 ± 2.08% in the ISP. A single ISP ganglion of the ileum contained on average 3.39 ± 1.60 P2X2-positive/galanin-positive nerve cells, and their mean diameter was 13.60 ± 3.11 µm. The percentages of P2X2-IR/galanin-IR neurons in the MP, OSP, and ISP of the sections of the small intestine (duodenum, jejunum, and ileum) are presented in Figure 4.

The statistical analysis showed no statistically significant differences (*p* < 0.05) in the mean number of P2X2-IR/galanin-IR neurons (*n* = 5) per ganglion in plexuses of the same type (MP vs. MP, OSP vs. OSP, ISP vs. ISP) between sections of the small intestine, i.e., the duodenum, jejunum, and ileum. Moreover, no statistically significant differences (*p* < 0.05) were found in the mean diameters of P2X2-positive/galanin-positive neurons in individual plexuses of the MP, OSP, and ISP between the duodenum, jejunum, and ileum. Additionally, no statistically significant differences (*p* < 0.05) were found between the number of P2X2-IR/galanin-IR neurons (*n* = 5) in the MP of the duodenum, jejunum, and ileum (Figure 4). However, there were statistically significant differences (*p* < 0.05) in the mean number of P2X2-IR/galanin-IR neurons (*n* = 5) in nerve plexuses of the same type (OSP vs. OSP, ISP vs. ISP) between sections of the small intestine (i.e., duodenum, jejunum, and ileum; see Figure 4.

### 3.3. P2X2/sP

Microscopic analysis of the MP, OSP, and ISP of the duodenum, jejunum, and ileum revealed the presence of P2X2-positive and sP-positive neurons. However, co-immunoexpression of P2X2 and sP was not detected in any ganglia (Figure 3D).

### 3.4. P2X2/VIP

In the MP of the duodenum, simultaneous immunoexpression of P2X2 and VIP was found in 41.4 ± 4.97% of nerve cells, which had a mean diameter of 17.65 ± 3.85 µm. A single MP ganglion of the duodenum contained 2.60 ± 1.35 P2X2-IR/VIP-IR neurons, while an OSP ganglion contained 3.00 ± 1.62 P2X2-positive/VIP-positive neurons, with a mean diameter of 14.42 ± 2.86 µm. The percentage of P2X2-IR/VIP-IR neurons in the OSP of the duodenum was 74.69 ± 4.10%. In the ISP of the duodenum, 51.21 ± 2.31% of nerve cells were P2X2-positive/VIP-positive, with a mean diameter of 12.93 ± 3.78 µm. A single ISP ganglion of the duodenum contained 3.90 ± 1.82 of P2X2-IR/VIP-IR neurons.

The presence of P2X2-IR/VIP-IR neurons was found in 42.35 ± 2.59% of nerve cells of the MP of the jejunum (Figure 3E). A single ganglion of this plexus contained 2.88 ± 1.75 P2X2 positive/VIP-positive neurons, with a mean diameter of 16.97 ± 2.63 µm. The OSP of the jejunum contained 50.08 ± 2.02% P2X2-IR/VIP-IR nerve cells, with a mean diameter of 14.55 ± 3.14 µm (Figure 3G). A single OSP ganglion of the jejunum contained on average 2.69 ± 1.34 neurons with simultaneous immunoexpression of P2X2 and VIP. A single ISP ganglion of the jejunum contained 2.65 ± 1.49 P2X2-IR/VIP-IR neurons, with an average diameter of 13.98 ± 3.44 µm (Figure 3F). The percentage of P2X2-positive/VIP-positive nerve cells in the jejunal ISP was estimated to be 51.38 ± 2.62%.

In the population of ileal MP nerve cells, 62.19 ± 3.27% were P2X2-IR/VIP-IR, with a mean diameter of 18.19 ± 3.30 µm. A single ganglion of this plexus contained on average 2.96 ± 1.31 P2X2-positive/VIP-positive neurons. In the OSP of the ileum, the average number of P2X2-IR/VIP-IR neurons per ganglion was 2.59 ± 1.55. The mean diameter of the OSP neurons in the ileum showing simultaneous P2X2 and VIP immunoreaction was 14.02 ± 2.45 µm. Among the OSP neurons of the ileum, 56.42 ± 2.30% showed P2X2 co-localization with VIP. The ISP of the ileum contained 49.54 ± 2.29% P2X2-IR/VIP-IR neurons, with a mean diameter estimated at 15.68 ± 2.85 µm. In a single ganglion of this plexus, the average number of neurons simultaneously immunoexpressing P2X2 and VIP was 3.31 ± 1.75.

No statistical differences (*p* < 0.05) were noted in the mean number (*n* = 5) of P2X2-IR/VIP-IR neurons per ganglion of a given type of plexus (MP vs. MP, OSP vs. OSP, ISP vs. ISP) between the duodenum, jejunum, and ileum. Moreover, no statistically significant differences (*p* < 0.05) were found between the mean (*n* = 5) diameters of P2X2-positive/VIP-positive nerve cells of plexuses of the same type (for MP, OSP, or ISP) in the duodenum, jejunum, and ileum. Comparison of the percentages (*n* = 5) of P2X-IR/VIP-IR neurons in the MP revealed statistically significant differences (*p* < 0.05) between the duodenum and the ileum and between the jejunum and ileum. In the OSP, statistically significant differences (*p* < 0.05) in the mean number (*n* = 5) of P2X2-positive/VIP-positive neurons were noted between all sections of the small intestine. No statistically significant differences (*p* < 0.05) were found in the mean (*n* = 5) number of neurons showing co-localization of P2X2 with VIP in the ISP of the duodenum, jejunum, and ileum. Data on the number of P2X2-IR/VIP-IR neurons in individual plexuses (MP, OSP, and ISP) of the sections of the small intestine are presented in Figure 5.

## 4. Discussion

The literature lacks data on the distribution of P2X2-IR neurons in the various ENS plexuses of the whole porcine small intestine, i.e., including all of its sections. This was analysed for the first time in this work, using double immunofluorescence staining, which identified P2X2-IR neurons in the MP, ISP, and OSP of the duodenum, jejunum, and ileum. It should be noted that the percentages of P2X2-IR neurons in the various ENS plexuses of the jejunum and ileum were comparable. In contrast, in the duodenum the number of P2X2-IR neurons (in the MP, ISP, and OSP) was statistically lower than in the other parts of the small intestine. Previous immunohistochemical studies have shown that the MP of the rat ileum contains approximately 70% P2X2-IR neurons [6], which corresponds to the results of our research, in which about 65% of ileal MP neurons were P2X2-positive. A very different degree of P2X2 receptor immunoexpression was found in a study conducted on the guinea pig ileum, in which only 25% of MP neurons were P2X2-IR [17]. Neurons located in the MP are divided into two types, S and AH, depending on their electrophysiology. AH neurons (mainly showing long-lasting action potential after hyperpolarization) are generally considered to be intrinsic primary afferent sensory neurons (IPANs) [19,20], whereas type S receives fast excitatory postsynaptic potentials (fEPSPs) and includes interneurons as well as motor neurons [19,20]. It has been demonstrated experimentally that ATP, which activates P2X receptors, takes part in the generation of fEPSPs in S neurons of the MP [21,22]. In the small intestine of P2X2 knockout mice, ATP did not depolarize S neurons but only AH neurons, whereas both types of neurons were depolarized in the group of mice with the P2X2 gene [18]. Moreover, that study showed that fEPSPs and peristalsis of the small intestine were inhibited in P2X2 gene knockout mice [18]. A slightly different experimental design (in the MP of the guinea pig small intestine) showed immunoexpression of P2X2 in calbindin-IR neurons (a marker of IPAN neurons), but also in nitric oxide synthase-IR (NOS) neurons, which is considered an indicator of inhibitory motor neurons and descending interneurons in this species [17,23]. Therefore, this neurochemical profile of neurons suggests the role of P2X2 receptors (activated by ATP) in the action of IPAN neurons as well as in the peristalsis of guinea pig intestines [17]. Additionally, previous findings indicate that about 56% of neurons in the SP of the rat ileum were P2X2-positive [6], compared to about 50% in guinea pigs [17]. The percentages of P2X2-IR neurons in the SP of the rat and guinea pig are generally comparable to the percentages of P2X2-positive neurons in the ISP and OSP of the pig small intestine (about 44–60%). However, it should be borne in mind that the SP of the rat and the guinea pig does not subdivide (in contrast to the pig), which is why these results can only partially be compared to our findings, which took into account the division of the SP. To summarize, the distribution of P2X2-IR neurons and their action in individual plexuses can be assumed to depend in part on the specific structure of the porcine ENS.

Galanin is a neuropeptide generally believed to perform a wide variety of functions in the digestive tract. However, it seems that its main function is its role in protecting various classes of neurons, including primary afferent neurons after axotomy [24]. The neuroprotective action of galanin is also suggested by the results of an experiment in pigs with acrylamide supplementation, in which galanin was overexpressed in the MP and SP neurons of the stomach [25]. Additionally, it is well known that galanin-IR neurons can be secretomotor or vasodilator neurons [26]. It is worth noting in this context that in the rat ileum the presence of P2X2 receptors was found to be quite common in secretomotor and vasodilator neurons of the ENS [27]. One of the best-known activities of galanin is its role in modulating digestive tract motility, indirectly by releasing other neurotransmitters or directly by acting on intestinal smooth muscle cells [28]. Galanin is also present in inhibitory motor neurons [26], whose activity is also regulated by ATP via P2X2 receptors [17]. Despite the presence of P2X2 receptors and galanin in the same types of neurons, their co-immunoexpression has thus far not been investigated. IHC staining of cryostatic sections showed statistically similar populations of P2X2-IR/galanin-IR neurons in the MP of the duodenum, jejunum, and ileum of the pig (from 20% to 25%). In addition, P2X2-IR/galanin-IR neurons in both the ISP and OSP were shown to oscillate between 25% and 35% of the nerve cells found in individual sections of the pig small intestine. Therefore, it can be assumed that galanin and P2X2 receptors probably have some (as yet undefined) functional relationship in the small intestinal ENS of the pig.

The analysis of IHC-stained preparations showed the presence of P2X2-positive/sP-negative neurons as well as P2X2-negative/sP-positive neurons; however, there was no simultaneous immunoexpression of P2X2 receptor and sP in any of the MP, ISP and OSP neurons of the porcine small intestine. A similar relationship, but with respect to the P2X3 receptor, was demonstrated in an IHC analysis of the guinea pig ileum, which also showed no co-localization of the receptors with sP [29]. Previous reports clearly indicate that in neonatal guinea pigs most MP P2X3-IR neurons also exhibit P2X2 immunoexpression [30]. This may be due to the fact that these two receptors form a heteromeric P2X2/3 receptor [31]. Based on the results of previous experiments, it can be presumed that the lack of sP co-localization with P2X2 may be linked to the functional diversity of ENS neuron types. Immunoexpression of P2X2 has previously been found mainly in inhibitory motor neurons, whereas sP was present in excitatory motor neurons [17,32]. It should be recalled that in this study, co-localization of P2X2 receptors was also shown with galanin, which may partly explain the lack of co-localization of P2X2 with sP, as galanin is believed to occur mainly in inhibitory motor neurons [26]. The results suggest that in the absence of the coexistence of sP and P2X2 receptors, sP has a negligible effect (if any) on purinergic signalling through P2X2 receptors in the small intestine of the pig. Nevertheless, more advanced functional tests are needed to verify this assumption.

Both VIP and ATP are believed to influence GIT motility by regulating the activity of inhibitory motor neurons [26]. In an IHC analysis of the guinea pig ileum, about 90% of SP neurons were shown to be P2X2-IR/VIP-IR [17]. For comparison, the present study found slightly smaller (from 50% to 70%) subpopulations of P2X2-IR/VIP-IR neurons. The presence of VIP in SP neurons (both in cell bodies and in their fibres supplying the mucosa) of the small intestine of mice determines the role of this neuropeptide as a neuroregulator of secretomotor processes [33]. This suggests that at least some of the P2X2-IR/VIP-IR nerve cells identified in the present study in the SP of the porcine small intestine may also play a role in regulation of secretomotor neurons. Moreover, P2X2-IR/VIP-IR nerve cells in the MP of the pig small intestine were shown to constitute a relatively high percentage of neurons (about 40–65%), which suggests a functional neuromodulatory effect of VIP with regard to purinergic transmission.

## 5. Conclusions

In conclusion, the present study sheds light on some aspects of purinergic signalling in the small intestine of the pig. For the first time, we found that P2X2-IR neurons are present in the MP, ISP, and OSP of all sections of the small intestine, which probably suggests that activation of this receptor may influence the function of the small intestine. We additionally concluded that there is most likely functional cooperation (in the ENS of the porcine small intestine) between the P2X2 receptor and the neuropeptides analysed (galanin and VIP), but not sP. Due to lack of P2X2 receptors selective agonist it is very difficult to predict exact P2X2 receptor’s role in the porcine small intestine. Future experimental work regarding the use of galanin and/or VIP knockout animals would throw a new light on the possible effects of these neuropeptides in ATP-related nociceptive signalling.

## Figures and Tables

**Figure 1 animals-12-03576-f001:**
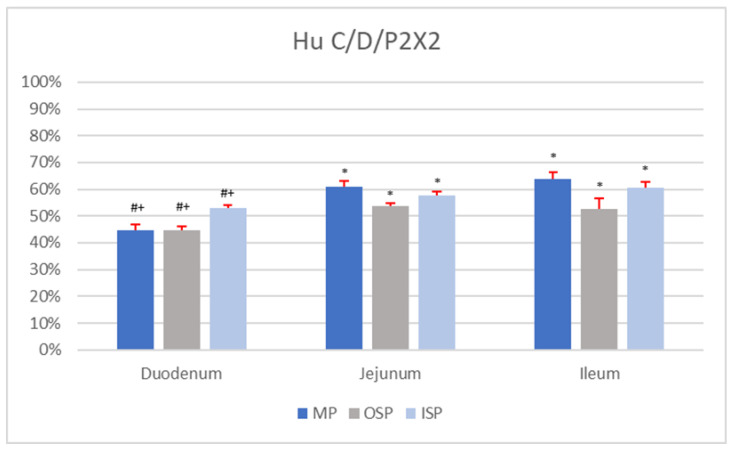
Percentage of Hu C/D-IR/P2X2-IR nerve cells in individual nerve plexuses (MP, OSP, and ISP) of the duodenum, jejunum, and ileum. Statistically significant differences (*p* < 0.05) between the mean number of Hu C/D-IR/P2X2-IR neurons present in analogous nerve plexuses of different sections of the small intestine are marked: * vs. duodenum, # vs. jejunum, + vs. ileum.

**Figure 2 animals-12-03576-f002:**
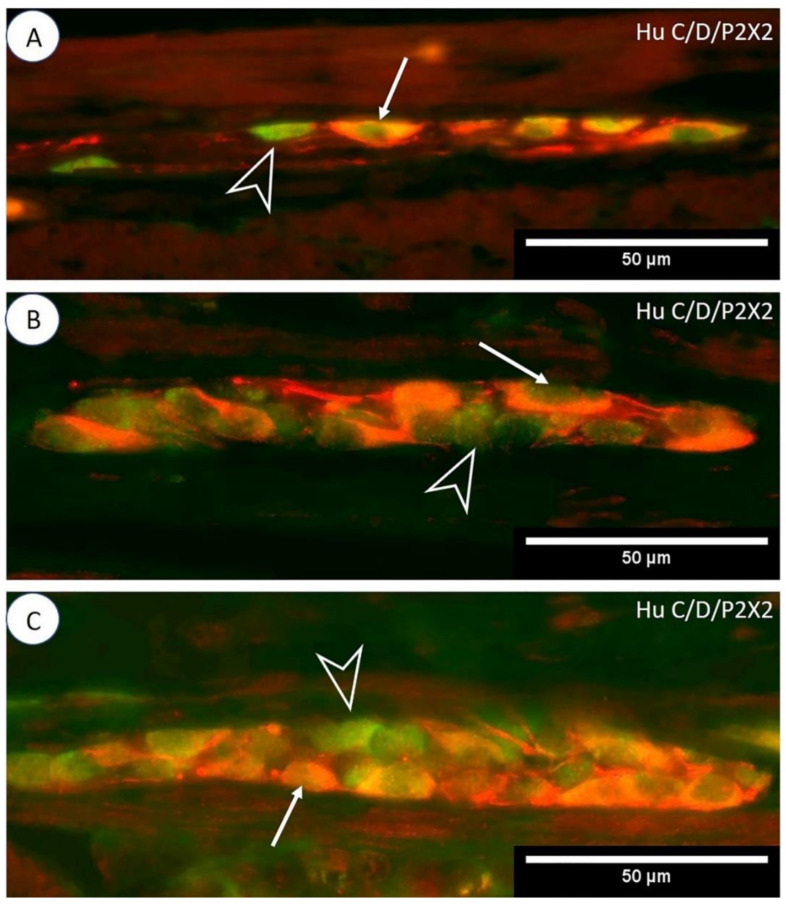
Immunoexpression of P2X2 receptors in porcine jejunum. Photos (**A**–**C**) show P2X2 double immunofluorescence staining with a Hu C/D neuronal marker in MP, OSP, and ISP ganglia (respectively). The arrow indicates Hu C/D-IR/P2X2-IR neurons, and the empty arrowhead indicates Hu C/D-positive/P2X2-negative neurons.

**Figure 3 animals-12-03576-f003:**
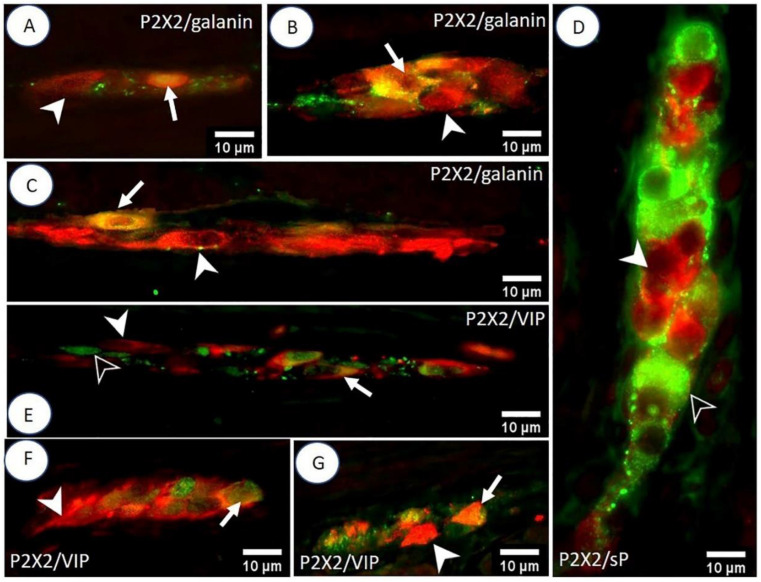
Double IHC staining of the porcine small intestine for P2X2 receptors and selected biologically active substances. The photos (**A**–**C**) show the MP, ISP, and OSP (respectively) of the duodenum. The arrow indicates P2X2-IR/galanin-IR neurons and the arrowhead indicates P2X2-positive/galanin-negative neurons. Micrograph (**D**) shows the lack of P2X2 co-localization with sP in the OSP of the ileum. The arrowhead indicates a P2X2-positive/sP-negative neuron, and the empty arrowhead indicates a P2X2-negative/sP-positive neuron. Photos (**E**–**G**) illustrate P2X2/VIP immunoreactivity in the MP, ISP, and OSP (respectively) of the jejunum. The arrow indicates P2X2-IR/VIP-IR neurons, the arrowhead indicates P2X2-positive/VIP-negative neurons, and the empty arrowhead indicates a P2X2-negative/VIP-positive neuron.

**Figure 4 animals-12-03576-f004:**
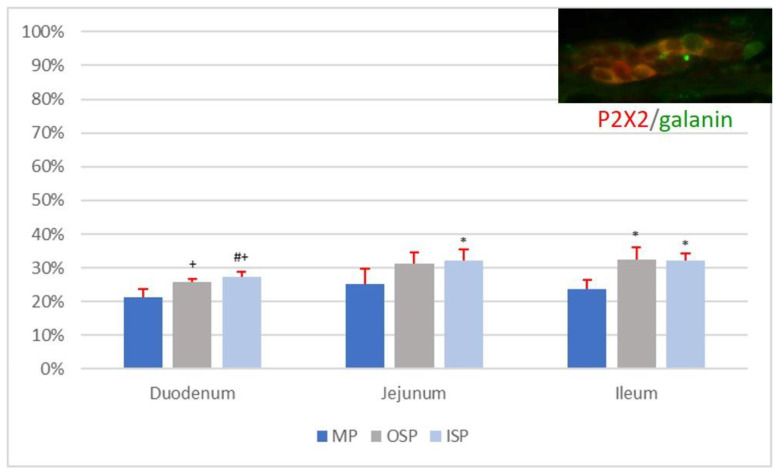
Percentage of P2X2-IR/galanin-IR neurons in individual nerve plexuses (MP, OSP, and ISP) of the duodenum, jejunum, and ileum. Statistically significant differences (*p* < 0.05) between mean numbers of P2X2-IR/galanin-IR neurons present in analogous nerve plexuses of different sections of the small intestine are marked: * vs. duodenum, # vs. jejunum, + vs. ileum.

**Figure 5 animals-12-03576-f005:**
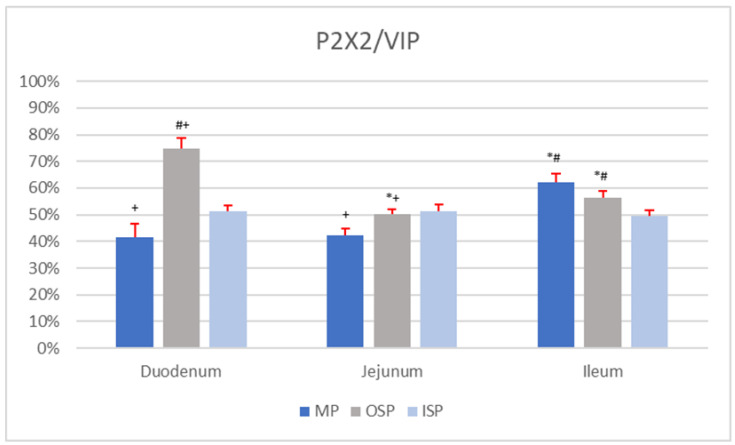
Percentage of P2X2-IR/VIP-IR neurons in individual nerve plexuses (MP, OSP, and ISP) of the duodenum, jejunum, and ileum. Statistically significant differences (*p* < 0.05) between mean numbers of P2X2-IR/VIP-IR neurons in analogous nerve plexuses of different sections of the small intestine were marked: * vs. duodenum, # vs. jejunum, + vs. ileum.

**Table 1 animals-12-03576-t001:** The detailed specification of used primary and secondary antibodies.

Antigen	Host	Dilution	Code	Source
P2X2	Rabbit	1:200	ab28469	Abcam, Cambridge, UK
Hu C/D	Mouse	1:400	A-21271	Thermo Fisher Scientific, Waltham, MA, USA
sP	Rat	1:200	8450-0505	BIO-RAD, Hercules, CA, USA
Galanin	Guinea pig	1:300	T-5036	BMA Biomedicals, Augst, Switzerland
VIP	Mouse	1:100	9535-0504	Biogenesis Inc., London, UK
Anti-mouse Alexa Fluor 488	Goat	1:800	A-11029	Thermo Fisher Scientific, Waltham, MA, USA
Anti-rabbit Aleksa Fluor 594	Donkey	1:800	A-21207	Thermo Fisher Scientific, Waltham, MA, USA
Anti-rat Alexa Fluor 488	Goat	1:800	A-11006	Thermo Fisher Scientific, Waltham, MA, USA
Anti-guinea pig Alexa Fluor 488	Goat	1:800	A-11073	Thermo Fisher Scientific, Waltham, MA, USA

**Table 2 animals-12-03576-t002:** Morphometrical analysis of enteric neurons showing simultaneous IR to Hu C/D and P2X2 in different segments of the porcine small intestine (mean ± S.D.). Statistically significant differences (*p* < 0.05) between the different parameters of analogous nerve plexuses of different segments of the small intestine are marked: a vs. duodenum, b vs. jejunum, c vs. ileum.

Segment of the Small Intestine	Plexus	Single Ganglion Area (µm^2^)	Number of Neurons per Ganglion	Neuron Diameter (µm)
Duodenum	MP	2826.15 ± 149.87 ^bc^	4.26 ± 1.95	20.18 ± 3.11
	OSP	2285.57 ± 177.42	4.82 ± 1.87	14.37 ± 2.63 ^bc^
	ISP	1711.00 ± 103.82 ^c^	4.11 ± 1.26 ^b^	14.73 ± 2.22
Jejunum	MP	4643.7 ± 335.91 ^a^	4.40 ± 1.70	21.24 ± 1.10
	OSP	1970.24 ± 172.62	5.32 ± 2.25	17.91 ± 1.12 ^a^
	ISP	1953.9 ± 142.23	6.64 ± 1.65 ^a^	15.82 ± 1.04
Ileum	MP	4449.81 ± 98.85 ^a^	6.52 ± 2.31	20.62 ± 1.53
	OSP	2175.18 ± 476.44	6.71 ± 1.82	18.52 ± 1.30 ^a^
	ISP	2239.73 ± 391.07 ^a^	8.87 ± 2.13	15.03 ± 0.97

## Data Availability

Not applicable.

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
