# Peer review of "Immunodetection of P2X2 Receptor in Enteric Nervous System Neurons of the Small Intestine of Pigs"

_animals, 2022, doi:10.3390/ani12243576_

Round 1

Reviewer 1 Report

The present manuscript authored by Mozel and Arciszewski is intended to assess by means of double immunofluorescence staining the segmental extend of immunoexpression of P2X2 receptor (purinergic) in enteric neurons of the porcine small intestine. In general, the article is well written and developed applying the sound scientific methodology, being of great interest to the research area. The data are potential to be interesting, convincing and the statistical analysis is appropriate. The manuscript comprises all the necessary elements of scientific novelty. I recommend this article for publication after incorporating minor changes given in below.

Line 58 – please abbreviate “the gastrointestinal tract” to “GIT”

Line 60  – The species-specific differences in ENS structure of mammals is not complete. Please add some information that the presence of two interconnected but separate intrinsic submucous networks can be also found in others animals’ species (see Timmermans et al. (2001), Outer submucous plexus: An intrinsic nerve network involved in both secretory and motility processes in the intestine of large mammals and humans. Anat. Rec., 262: 71-78. for reference).

Line 56, 60, 64 – please remove “;” from the sentences.

Line 87 – anatomically there is the small intestine and large intestine. Thus, the term “intestines” is not clear. Maybe it would be better to replace it with “the gut”.

Line 91 – “fixed” instead of “fixated”.

Line 94, 96  – what kind of tissues? Histologically, there are four types. I think it would be more appropriate to refer to the whole organ and replace it with “samples” or “intestine samples”.

Line 110 – Please check the correctness of this sentence. I think antibodies against P2X2 were mixed with either VIP, substance P or galanin.

Line 110, 396 – The term “co-expression” suggests that the authors refers to genes. But in the following part it become apparent that they referred to immunohistochemical studies. Any sentences where the term “expression” is used in wrong context of protein production/secretion must be rephrased.

Line 119 – “antibodies” but not a single antibody!

Line 140, 161 – “immunoreactive” was already abbreviated in line 75.

Table 1 – what is “Szwajcaria”???

Line 151 – neurons cannot be “with colocalization”. Co-localization of two substances may be found/present in neurons. This is kind of condition but not possession. Please change to “neurons co-localizing …”, “neurons showing co-localization of …” or something similar.

Line 153 – “segment” instead of “section”.

Line 157 – please change to “µm2”.

Line 203 – the authors should explain what they meant by “plexus type”? The main difference between myenteric and submucosal plexuses are their localization, but not type. I think the authors meant “corresponding plexuses (MP vs. MP ….)”.

Line 230, 339, 342 – “open arrowheads” should be replaced with “empty arrowhead” or “hollow arrowhead”.

Reference  nr. 22 - please check the spelling of the authors' names

Author Response

Reviewer 1

General comment - In general, the article is well written and developed applying the sound scientific methodology, being of great interest to the research area. The data are potential to be interesting, convincing and the statistical analysis is appropriate. The manuscript comprises all the necessary elements of scientific novelty. I recommend this article for publication after incorporating minor changes given in below.

Thank you very much for your questions and suggestions! We have included essential information, additional explanations and revised expressions in the corrected Manuscript. Please refer to the following point-to-point response. We hope that we have answered all your questions. Thanks for your time.

Comment – please abbreviate “the gastrointestinal tract” to “GIT”

Thank you for pointing this out. Gastrointestinal tract has been abbreviated to GIT. This abbreviation is further used.

Comment - The species-specific differences in ENS structure of mammals is not complete. Please add some information that the presence of two interconnected but separate intrinsic submucous networks can be also found in others animals’ species (see Timmermans et al. (2001), Outer submucous plexus: An intrinsic nerve network involved in both secretory and motility processes in the intestine of large mammals and humans. Anat. Rec., 262: 71-78. for reference).

We thank the Reviewer for pointing out that we have not elaborated enough in the Introduction on the organization of ENS in large animal species. In the updated Introduction section in the revised manuscript we have expanded the information about this issue and have included the proposed reference.

Comment – please remove “;” from the sentences.

Done.

Comment - anatomically there is the small intestine and large intestine. Thus, the term “intestines” is not clear. Maybe it would be better to replace it with “the gut”.

We totally agree with the Reviewer. The term “gut” was used instead of “intestines”.

Comment - “fixed” instead of “fixated”.

We fully agree. Corrected.

Comment - what kind of tissues? Histologically, there are four types. I think it would be more appropriate to refer to the whole organ and replace it with “samples” or “intestine samples”.

Thank you for pointing this out. We have corrected this as suggested.

Comment - Please check the correctness of this sentence. I think antibodies against P2X2 were mixed with either VIP, substance P or galanin.

Once again thank you for this correction. Indeed, the construction of this sentence was awkward. We have corrected it to “For co-localization studies P2X2 antibodies were combined with either VIP, galanin or sP antibodies. “

Comment - The term “co-expression” suggests that the authors refers to genes. But in the following part it become apparent that they referred to immunohistochemical studies. Any sentences where the term “expression” is used in wrong context of protein production/secretion must be rephrased.

We fully agree with the Reviewer. Throughout the text, term “co-expression” was replaced with “co-immunoexpression”.

Comment - “antibodies” but not a single antibody!

Indeed, we used antibodies raised against antigens but not a single antibody. Corrected as requested.

Comment - “immunoreactive” was already abbreviated in line 75.

Thank you for pointing this out. Corrected throughout the text.

Comment - what is “Szwajcaria”???

We apologize for this embarrassing mistake. Corrected to “Switzerland”

Comment - neurons cannot be “with colocalization”. Co-localization of two substances may be found/present in neurons. This is kind of condition but not possession. Please change to “neurons co-localizing …”, “neurons showing co-localization of …” or something similar.

We absolutely agree with the Reviewer. Co-localization in relation to neurons means that at least two different substances (possibly neurotransmitters or neuromodulators) are present in the same nervous cell at the same time. Of course, we have corrected this ambiguous expression as suggested by the Reviewer.

Comment - “segment” instead of “section”.

Thank you for pointing this out. Corrected as requested.

Comment - please change to “µm2”.

We apologize for this obvious typo. Corrected throughout the text.

Comment - the authors should explain what they meant by “plexus type”? The main difference between myenteric and submucosal plexuses are their localization, but not type. I think the authors meant “corresponding plexuses (MP vs. MP ….)”.

We totally agree with the Reviewer that the expression “plexus type” is awkward. There are no plexus types in ENS. We changed this expression to “analogous plexuses” as suggested. 

Comment - “open arrowheads” should be replaced with “empty arrowhead” or “hollow arrowhead”.

Thank you again for this language correction. Done as requested.

Comment - Reference  nr. 22 - please check the spelling of the authors' names.

Checked and corrected.

Reviewer 2 Report

Manuscript ID: animals-2079330

Title: Immunodetection of P2X2 receptor in enteric nervous system neurons of

the small intestine of pigs

In this manuscript, the authors investigated the presence of the P2X2 receptor in the neurons of the enteric nervous system (ENS) of the pig small intestine (duodenum, jejunum, and ileum) by the immunofluorescence method.It’s a very interesting study. Only few questions should be addressed to make it optimized.

1.      Why choose galanin-immunoreactive (IR), substance P-IR (sP), and vasoactive intestinal polypeptide (VIP-IR) neurons to innvestigate the relationship of P2X2?

2.      How to separation out MP, OSP, and ISP, then proceeding immunofluorescence? Or how to proceed immunofluorescence on MP, OSP, and ISP, respectively?

3.      Figure 1, 3, and 5 should add the images were captured in the stained sections, to more intuitive presentation of results.

4.      The results of single ganglion area, number of neurons per ganglion, and neuron diameter should list in a table.

5.      Line 335-340 14A-14C, 14D, and 14E-14G is Figure 5A-5C, 5D and 5E-G, respectively?

6.      In discussion section, should add some significance or speculation to the theory of the field.

Author Response

Reviewer 2

General comment - In this manuscript, the authors investigated the presence of the P2X2 receptor in the neurons of the enteric nervous system (ENS) of the pig small intestine (duodenum, jejunum, and ileum) by the immunofluorescence method.It’s a very interesting study. Only few questions should be addressed to make it optimized.

Thank you very much for your questions and suggestions! We have included essential information, additional explanations and revised expressions of the revised Manuscript. Please refer to the following point-to-point response. We hope that we have answered all your questions. Thanks for your time!

Comment 1.      Why choose galanin-immunoreactive (IR), substance P-IR (sP), and vasoactive intestinal polypeptide (VIP-IR) neurons to innvestigate the relationship of P2X2?

We greatly appreciate this reviewer’s comment. In order to eliminate the reviewer’s concern on this issue, we gave more explanation as seen below.

“Therefore the present study aimed to determine the degree of immunoexpression of the P2X2 receptor and to demonstrate whether the P2X2 receptor is associated with galanin-immunoreactive (IR), substance P-IR (sP), and vasoactive intestinal polypeptide (VIP-IR) neurons in the ENS of the small intestine of pigs. The selection of galanin, sP and VIP was based on the previous findings showing their neurotransmittery role in certain subpopulations of enteric neurons (including sensory and secretomotor) as well as potential neuroprotective role in the gut disorders with the pain symptoms.”

Comment 2.      How to separation out MP, OSP, and ISP, then proceeding immunofluorescence? Or how to proceed immunofluorescence on MP, OSP, and ISP, respectively?

We appreciate the Reviewer’s suggestion. However, we would like to explain that we did not work with whole-mount preparations (which are the best option to separate the myenteric plexus from layers of the submucosal plexus). We used instead, cryostat sections which are also routinely used to identify ganglia/neurons of MP, OSP, and ISP. Moreover, in the pig small intestine the distances between OSP an ISP are large enough to make any necessary judgement and calculations. Because of thickness of muscular layer as well as submucosal layer mentioned ganglia were easily identified. Additionally, immunofluorescence stainings of separated ganglionic plexuses would require larger amounts of expensive antibodies which is a significant economic issue, especially in light of our low research budget.

Comment 3.      Figure 1, 3, and 5 should add the images were captured in the stained sections, to more intuitive presentation of results.

We are grateful the Reviewer for this comment. The idea of figure 1, 3 and 4 was to present numerical statistical data in a graph form. This explains why (as a rule) the mentioned figures did not contain any images. Please note, that corresponding images can be found in figure 2 and 5. However, we agree with the Reviewer that the perception of figures might be not informative enough so we decided to add explanatory subheadings like “P2X2 / Hu C/D” in figure 1 and 2, “P2X2 / galanin” in figure 3 and 5 as well as “P2X2 /VIP” in figure 4 and 5. We hope, this correction is acceptable for the Rewiever.

Comment 4.      The results of single ganglion area, number of neurons per ganglion, and neuron diameter should list in a table.

Thank you for this suggestion, we fully agree with that.  In revised version of the manuscript you can find a requested table.

Comment 5.      Line 335-340 14A-14C, 14D, and 14E-14G is Figure 5A-5C, 5D and 5E-G, respectively?

Thank you for pointing this out. We have corrected this as suggested.

Comment 6.      In discussion section, should add some significance or speculation to the theory of the field.

We greatly appreciate this reviewer’s comment. Although we tried to outline the significante of our study in Conclusion section we decided to add some perspectives for future research in the field. In order to eliminate the reviewer’s concern on this issue, we gave more explanation as seen below.

“Due to lack of P2X2 receptors selective agonists it is very difficult to predict the exact P2X2 receptors role in the porcine small intestine. Future experimental work regarding the use of galanin or VIP knockout animals would through a new light on the possible effects of these peptides in ATP-related nociceptive signaling.”

Round 2

Reviewer 2 Report

In revised manuscript,  the authors responsed is very well. But I didn't found  the images were captured in the stained sections of Figure 3 , which could make the results more credible. So I still suggests authors add it.

  •  

Author Response

As we fully respect the expertise of the Reviewer we decided to correct the Figure 3 according to the suggestion.